# Characterizing rurality using the *All of Us* Research Program data

**Michael Bradfield[1], Toluwanimi Olorunnisola(iD)[2]\*, Vignesh Subbian[2]**

**1** Department of Family Medicine, Banner Health North Colorado Medical Center, Greeley, Colorado, United States of America, **2** College of Engineering, The University of Arizona, Tucson, Arizona, United States of America

\* taolorunnisola@arizona.edu

## Abstract

Rural communities experience disproportionately higher rates of chronic diseases, less access to healthcare services, and poorer health outcomes compared to their urban counterparts in the United States. However, inconsistencies in how rurality is defined across biomedical research, including limitations in geographic detail within large-scale datasets, present significant challenges for reliably studying rural health outcomes. This study aimed to develop and apply an operational rurality scale using 3-digit ZIP codes to characterize rural participation in the *All of Us* Research Program and to examine associations between rurality, delayed care, and healthcare affordability. Publicly available information from the Federal Office of Rural Health Policy and the Environmental Systems Research Institute was integrated to generate a continuous rurality scale at the 3-digit ZIP code level. A Kolmogorov-Smirnov test identified statistically significant differences in the geographic distribution of those who had delayed access to care (P < 0.001) and those with difficulties affording care (P < 0.001). The proposed continuous rurality scale is reproducible and extensible in several ways within the *All of Us* Workbench, as it provides a framework for categorizing participants by geolocation and facilitates standardized analyses of rurality-related research questions.

## Introduction

Rural communities experience disproportionately higher rates of chronic diseases, less access to healthcare services, and poorer health outcomes compared to their urban counterparts in the U.S. [1]. Moreover, there is a widening rural-urban divide in life expectancy in the U.S., with rural residents experiencing stagnant or declining lifespans amid urban gains, which is largely driven by cardiovascular disease mortality and working-age deaths [2,3]. For instance, between 1990 and 2017, life expectancy at birth increased by 3.9 years in metropolitan areas but only 0.6 years in nonmetropolitan areas [3].

**Data availability statement:** The data used in this study were obtained from the National Institutes of Health (NIH) All of Us Research Program, which is a third-party repository. As per the program's Data and Statistics Dissemination Policy (https://support.researchallofus.org/hc/en-us/articles/22346276580372-Data-and-Statistics-Dissemination-Policy), registered users are prohibited from exporting or publicly sharing participant-level information. As a result, the underlying dataset cannot be deposited to public repositories such as Figshare or Dryad. Researchers who meet the criteria for access to confidential data may register and request access to all reproducible artifacts through the All of Us Researcher Workbench (https://www.researchallofus.org/data-tools/workbench/). Aggregate data are publicly available on the All of Us Research Hub (https://databrowser.researchallofus.org/). Summary statistics that comply with dissemination policies are included in the manuscript.

**Funding:** The All of Us Research Program is supported by the National Institutes of Health, Office of the Director, including grants OT2OD026549 and OT2OD036485. There was no additional external funding received for this study.

**Competing interests:** The authors have declared that no competing interests exist.

Studying rural health is challenging, in part due to the many definitions of "rurality" among researchers, healthcare entities, and the federal government. The notion of rurality encompasses many ideological, demographic, economic, and cultural constructs, though most acknowledge that the definition should include some geographic component [4–7]. Definitions and geographic units used to determine rurality vary widely across biomedical and health services research. These inconsistencies influence not only research findings but also policy and how health disparities are identified and resources allocated [8].

Depending on the source, approximately one in five Americans lives in rural areas [5,6,9]. Rural health is adversely affected by many factors, including geographic isolation, inadequate healthcare infrastructure, physician shortages, poverty, low educational attainment, poor health literacy, and inadequate public transportation [4,10,11]. Overall, this culminates in an elevated disease burden and decreased life expectancy for rural populations [11]. One of the first steps in addressing these discrepancies is a fully operational and contextually relevant measure of rurality [12]. This is especially important for research using large-scale, real-world health datasets such as the *All of Us* Research program [13], where geographic identifiers are typically restricted to 3-digit ZIP codes to protect participant privacy. Without a consistent and privacy-preserving method for defining rurality, efforts to examine rural health disparities and link geographic context to health outcomes remain limited in scope and impact.

The National Institute of Health's *All of Us* Research Program is a precision medicine initiative that aims to enroll one million or more American participants [13,14]. One of the primary goals of the program is to engage with and reduce health disparities among traditionally marginalized groups, including those who are geographically underserved in biomedical research [14]. The *All of Us* data holds significant value when studying rural health outcomes, as it includes a more diverse range of data types and sources than most existing datasets, including demographic data, geo-location data, survey responses, electronic health record (EHR) data, and genomic data [15].

The *All of Us* Research Program specifies that residents of established rural and non-metropolitan ZIP codes that meet the Health Resources and Services Administration (HRSA) Federal Office of Rural Health Policy (FORHP) rural grant eligibility criteria are underrepresented in biomedical research [16–18]. The FORHP definition is broad and includes all non-metropolitan counties, certain commuting areas, and low-population density areas [16]. Despite adopting the FORHP definition, the *All of Us* program does not provide any readily available method or indicator to identify and classify participants based on rurality within the dataset.

Therefore, the purpose of this study was to develop and apply a rurality scale based on 3-digit ZIP codes to identify and characterize rural participation and enrollment within the *All of Us* Research Program. We then applied this scale to examine patterns in healthcare access and utilization, with the goal of informing future research on rural health disparities and enhancing the utility of large-scale datasets for rural health equity research, policy making, and health advocacy.

## Methods

### *All of Us* Program methods for recruitment and data collection

The *All of Us* Research Program recruits participants through multiple mechanisms, including academic medical centers, healthcare provider organizations, community-based enrollment sites, and digital platforms. Upon enrollment, participants complete a comprehensive informed consent process, which includes consent for long-term participation, sharing of electronic health records (EHRs), completion of health-related surveys, biospecimen donation (blood, saliva, and urine), and return of research results. The consent process also includes education about data use, privacy protections, and the ability to withdraw at any time [19,20]. Participant data are then harmonized and standardized using the Observational Medical Outcomes Partnership (OMOP) Common Data Model [21]. To protect participant privacy, *All of Us* data undergo a series of transformations, including removal of personally identifiable information, before being made available to researchers in a secure, cloud-based Researcher Workbench [22]. This study was conducted under an approved Data Use Agreement and adheres to all ethical research conduct and data use policies established by the *All of Us* Research Program.

### *All of Us* Researcher Workbench

The *All of Us* Researcher Workbench operates under a data passport model, where authorized users can access data and execute research projects. The Workbench offers two primary tiers of data access that investigators can use for research purposes: a Registered Tier and a Controlled Tier. The Registered Tier consists of data from EHR, wearable devices, survey responses, and physical measurements. The Controlled Tier holds genomic data, including whole genome sequencing, genotyping arrays, and the first three-digit ZIP code geolocation data. The Workbench provides collaborative workspaces, an interactive Jupyter Notebook environment with the ability to perform analyses using R or Python, and tools for developing study cohorts. We conducted our analysis using Python 3 (version 3.10.12) and utilized the *All of Us* Controlled Tier workspace using the version 8 curated dataset released in February 2025. Data analysis was performed between March 2023 and February 2025.

### Creating a rurality scale using geolocation codes

The *All of Us* program provides three-digit ZIP codes as part of the Controlled Tier dataset [23], in accordance with the Health Insurance Portability and Accountability Act (HIPAA) privacy rule for de-identifying Protected Health Information [24]. However, restricting geographic detail to three-digit ZIP codes presents a methodological challenge for studying rural populations. To address this challenge, we constructed a rurality measure by analyzing two publicly available data sources: the Federal Office of Rural Health Policy (FORHP) dataset [16], which lists all rural U.S. ZIP codes and the Environmental Systems Research Institute (ESRI) dataset [25], which includes all ZIP codes nationally. We aggregated this information to generate a classification scheme for rural 3-digit ZIP codes, as illustrated in Fig 1.

First, we obtained all U.S. five-digit ZIP codes and their corresponding population from the U.S. ESRI ZIP code geodatabase. The list of rural five-digit ZIP codes from the FORHP dataset was then used to tag all ZIP codes as "rural" or "non-rural." Next, we grouped all five-digit ZIP codes based on their first three digits. For each three-digit ZIP code group, a rural percentage was computed by taking the ratio of the population of those codes marked as "rural" divided by the total population of that three-digit ZIP code group (Fig 1). Finally, each *All of Us* participant was mapped to a place in the rurality scale based on their corresponding three-digit ZIP code to enable further analysis.

### Healthcare access and utilization survey data preparation and analysis

In addition to geolocation codes (3-digit ZIP codes), we used the following data: demographics (age, sex, ethnicity, and race), educational status, and responses to the Healthcare Access and Utilization survey [26], which includes 114

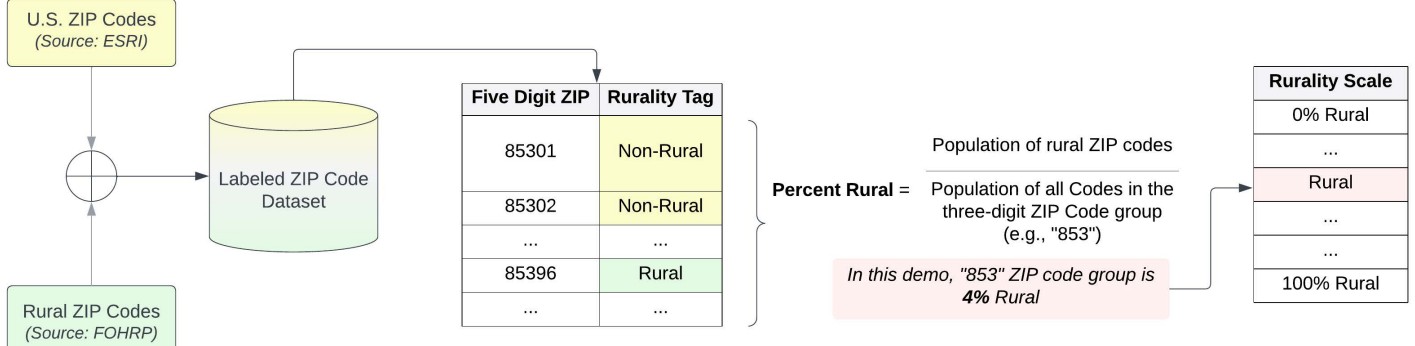

**Fig 1. Demonstration of developing a Rurality Scale with three-digit ZIP codes in a de-identified safe harbor dataset.**

questions [27]. We reviewed all 114 questions and extracted those questions relevant to delayed care (9 questions) and healthcare affordability (14 questions). Delayed care was assessed using nine survey items related to healthcare access; participants with six or more affirmative responses were coded as 1, indicating significant experiences with delayed care, and fewer than six were coded as 0. Healthcare affordability was evaluated using fourteen items, with nine or more affirmative responses coded as 1, and fewer than nine coded as 0. Conservative thresholds were selected through careful expert deliberations to reflect a high burden of barriers in each domain and to capture participants with persistent or widespread challenges rather than isolated instances. Fig 2 illustrates the coding process for classifying participants' responses.

## Statistical analysis of the survey data

We analyzed the survey data on delayed care and healthcare affordability by matching the rural percentages to each *All of Us* participant's corresponding ZIP codes and developed Empirical Cumulative Distribution Function (ECDF) plots to measure trends along the rural scale from 0% to 100% rural. The ECDF plot serves as a non-parametric method for visualizing the cumulative distribution of a dataset by displaying the cumulative probability associated with each data point. To compare and quantify statistical differences between the ECDF plots, we applied another non-parametric test, the Kolmogorov-Smirnov (KS) test. Additionally, we examined sociodemographic characteristics of participants classified in the 0% and 100% rural categories to assess differences between those residing in fully urban versus fully rural areas, as determined by their 3-digit ZIP code areas. These groups were selected to reflect clear, binary classifications of rurality and to minimize misclassification bias. Descriptive statistics, including frequencies and percentages for categorical variables and means with standard deviations for continuous variables, were generated for each group.

## Results

As of July 2025, the *All of Us* version 8 curated dataset had an overall sample size of 633,540 enrolled participants. We included all participants in the analysis. However, only a fraction of the total population (n = 305,860) completed the Healthcare Access and Utilization survey questions. The map in Fig 3 represents the percent rural result of each three-digit ZIP code of *All of Us* participants in the United States.

## Sociodemographic characteristics of the rural and non-rural cohort

We examined the sociodemographic characteristics of the 0% and 100% rural categories (Table 1). The 0% rural population comprised 57% of the participants (358,681 out of 633,540). Of these, 98% (354,841 out of 358,681) provided

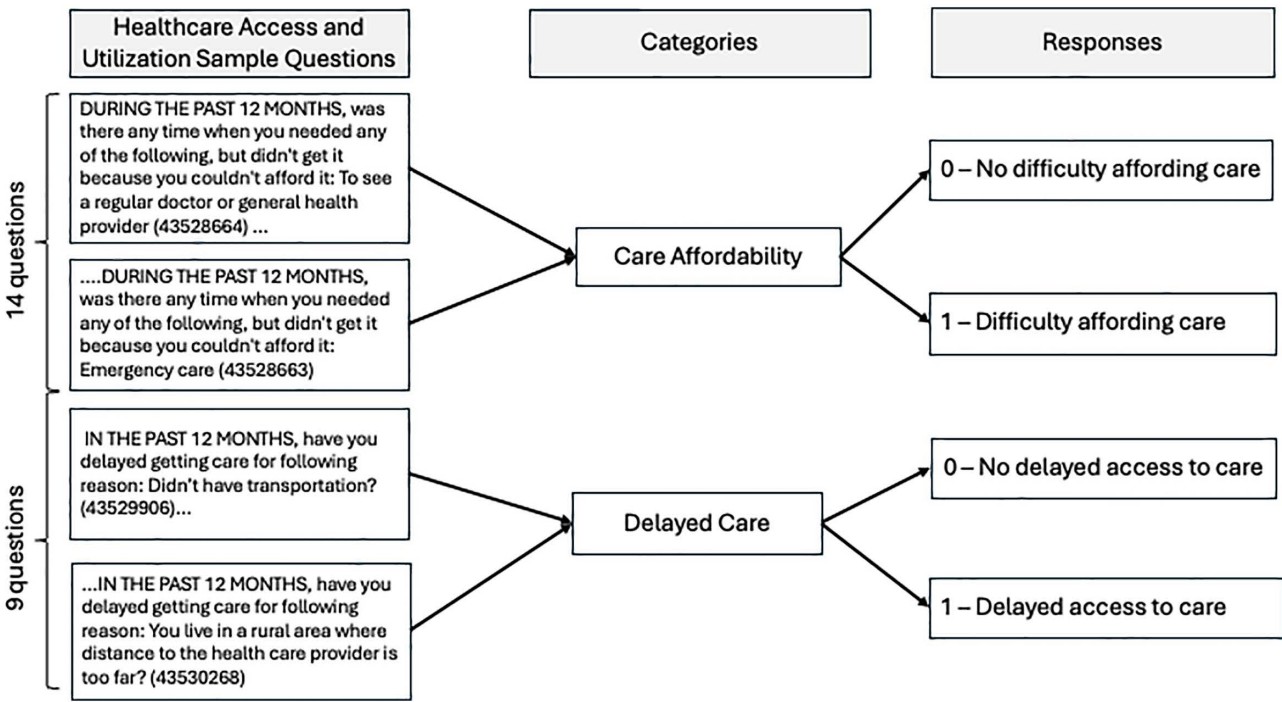

**Fig 2. An illustration of the coding process for Healthcare Access and Utilization survey questions.** Delayed care was assessed using nine (9) survey items, with participants having six (6) or more affirmative responses coded as 1 and fewer than six (6) coded as 0. Similarly, healthcare affordability was assessed using fourteen (14) survey items, with nine (9) or more affirmative responses coded as 1, and fewer than nine (9) coded as 0.

sociodemographic information. The average age was 54, with 50% identifying as White and 48% having a college or advanced degree.

The 100% rural cohort included 1.9% of enrolled participants (11,997 out of 633,540). Of these, 99% (11,929 out of 11,997) had sociodemographic information. The average age was 54, with 86% being white, 68% female, and 41% having a college education or advanced degree.

### Variations in healthcare access by geolocation

The ECDF plot (Fig 4) compares the distribution of access to care among each of the *All of Us* participants along the rural scale. A Kolmogorov-Smirnov test to compare the two distributions showed a statistically significant difference between those with and without delayed access to care (p < 0.001).

Similarly, as shown in Fig 5, participants reporting difficulty affording care tend to come from areas with slightly higher rural percentages than those with no difficulty affording care, and this difference is statistically significant (p < 0.001).

### Discussion

Variations in traditional definitions of rurality come from a whole host of considerations, including geographic remoteness, demographics, resources, communities' perceptions of their own rurality, population density, commuting patterns, and proximity to large urban areas [4,6,7,28]. All rural taxonomies incorporate some aspect of geography, and thus any consideration of the operationalization of "rurality" in a program, such as the *All of Us* Research Program, should start with the geographic grouping of participants in a systematic and reproducible manner.

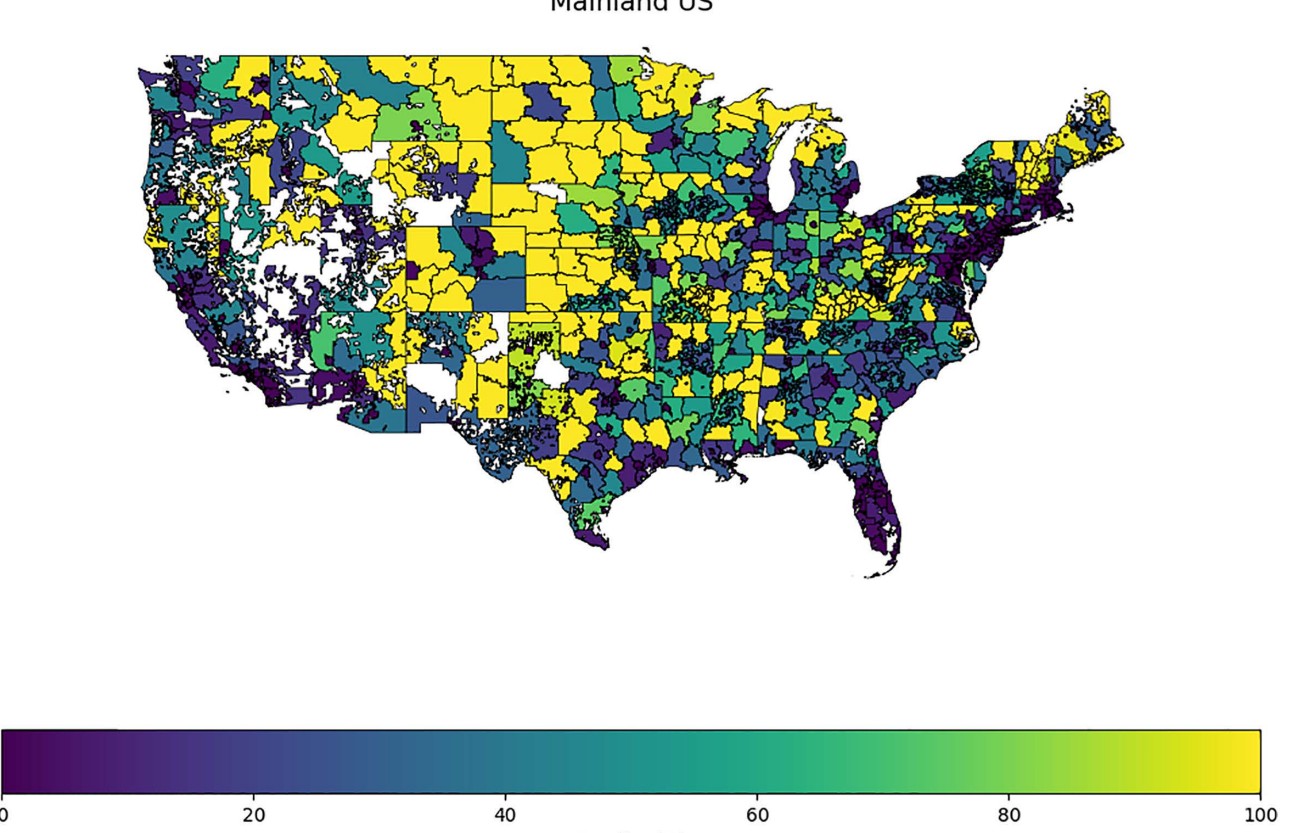

**Fig 3. Map illustrating the distribution of three-digit ZIP codes by percent rural among All of Us participants in the United States.** Note: The map was created by the authors using publicly available ZIP Code data from the U.S. Census Bureau (available at https://www.census.gov/geographies/mapping-files/time-series/geo/cartographic-boundary.html) and the All of Us Research Program participant data. In accordance with the program policy, this work acknowledges the essential contributions of All of Us participants.

Data from the *All of Us* research program holds significant potential for advancing the understanding of health outcomes of people living in rural America [13]. The diversity of participants, combined with the breadth of data available in the *All of Us* program, provides a unique opportunity to explore rurality through multiple lenses, including genomics, electronic health records (EHR), and survey responses. However, to fully leverage this potential, a formal, relevant, and operational rurality scale is essential [28]. A key methodological strength of our work is the application of a continuous rurality scale that is compatible with de-identified data constraints common in large, real-world health datasets. Most available resources limit geographic identifiers to 3-digit ZIP codes to protect participant privacy, making traditional, more granular rural classifications impractical. Our approach addresses this gap and offers a practical solution for researchers working within similar data environments.

A close examination of Table 1 shows the discrepancy between enrollment numbers in rural and urban locations within the *All of Us* Research Program. The number of rural participants with complete geolocation, EHR, genomic, and survey response data available is even more limited. Furthermore, Figs 4 and 5 show that participants living in areas with higher rurality percentages, as defined by our rurality scale, report statistically significant disparities in delayed care and healthcare affordability compared to their urban counterparts.

**Table 1. Comparison of the sociodemographic characteristics between 0% rural and 100% rural\*.**

| Characteristics | 0% rural | 100% rural |
|---|---|---|
| | **n = 354,841** | **n = 11,929** |
| **Race, n (%)** | | |
| White | 177,594 (50.0) | 10,321 (86.5) |
| Black/African American | 74,696 (21.1) | 565 (4.7) |
| Others or more than one | 98,146 (27.7) | 864 (7.2) |
| Missing | 4,405 (1.2) | 179 (1.5) |
| **Ethnicity, n (%)** | | |
| Non-Hispanic | 270122 (76.1) | 10,958 (91.9) |
| Hispanic | 73,821 (20.8) | 602 (5.0) |
| Others | 6,493 (1.8) | 190 (1.6) |
| Missing | 4,405 (1.2) | 179 (1.5) |
| **Sex assigned at birth, n (%)** | | |
| Female | 216,947 (61.1) | 8139 (68.2) |
| Male | 134,014 (37.8) | 3699 (31.0) |
| Others | 3,880 (1.1) | 91 (0.8) |
| **Age, mean (SD)** | 54 (17) | 54 (7.3) |
| **Education, n (%)** | | |
| College/Advanced Degree | 169,094 (47.7) | 4920 (41.2) |
| Less than College Graduate | 114,196 (32.2) | 4578 (38.4) |
| High School Graduate | 61,886 (17.4) | 2239 (18.8) |
| No Formal Education/Missing | 9665 (2.7) | 192 (1.6) |

Strengthening rural health research within *All of Us* will require both the development of a consistent rural taxonomy and continued efforts to enhance rural participation and retention across diverse geographic regions. As an extension to our study, the three-digit ZIP code classification system (i.e., the rurality scale) can be triangulated with survey response questions, EHR data, and other information within the Researcher Workbench to define more granular cohorts for answering important rurality-related research questions. Replicating the proposed rurality scale in the *All of Us* Researcher Workbench allows for characterizing the current rural participation and the associated demographic characteristics. Of note, the inverse of this scale also presents a method to study urban populations and is a first step in realizing the abilities of geolocation function within the *All of Us* Research Program.

When comparing our rurality scale to existing measures, such as the Rural-Urban Commuting Area (RUCA) codes [29], and the Index of Relative Rurality (IRR) [30], our rurality scale offers a standardized classification system that is compatible with three-digit ZIP codes. It provides a strong starting point for researchers studying rurality in de-identified large-scale datasets such as the *All of Us* Research Program, as it enables triangulation with survey questions, EHR data, and other factors to systematically group participants by rurality. The *All of Us* Research Program data provides more depth than many other databases in terms of its ability to capture demographic, social, genomic, clinical, and geographic characteristics from a very large cohort [13,23]. Creating ways to study rural health outcomes within the Researcher Workbench will add value to rural health research. Importantly, the program's focus on the underserved in biomedical research could provide novel insights into the health and behaviors of the most vulnerable people living in rural areas [14,31].

## Limitations and future work

Our study has several important limitations. The development of rural taxonomies requires consideration of as many factors of rurality as possible, including population size, remoteness, rural self-identification, commuting patterns, and

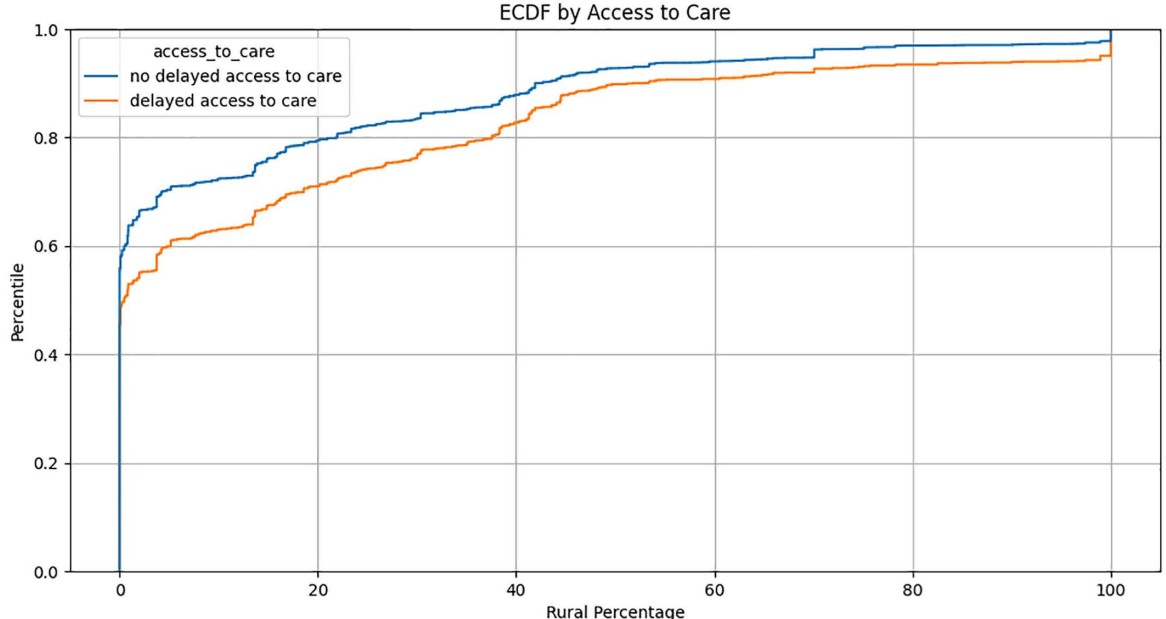

**Fig 4. The ECDF Plot shows the cumulative distribution of Access to Care along the rurality scale.** At 0% rural, we observe that the two curves uniformly rise steeply, indicating that a significant proportion of the participants have similar levels of access to care. Subsequently, the two curves split at different percentiles, indicating a disparity in access to care with respect to the rural percentage. The orange curve (delayed access to care) indicates that as the percentile increases, those experiencing delayed access to care tend to come from areas with higher rural percentages when compared to those with no delayed access to care (blue curve).

proximity to urban areas. In the *All of Us* Research Program, the basis for geographic parameters is defined as the three-digit ZIP code based on the FORPH datasets. The limited precision of three-digit ZIP codes may lead to misclassification by masking local heterogeneity and factors such as remoteness and travel distance to health resources.

A pre-determined geographic unit is prevalent in large-scale studies and government entities because most programs are constrained by funds, data availability, and participation. As with other studies utilizing large, real-world health data-sets, the methods applied in this work are driven by the requirement to protect participant privacy. Our approach was designed to meet these privacy standards while providing a practical, scalable method to characterize rurality within the constraints of available geographic information.

While some of our analyses focused on participants in the 0% and 100% rural categories, the continuous rurality scale developed here is intended for broader use. Further research is needed to establish empirically driven thresholds along the scale to differentiate urban, suburban, and rural areas, and to explore potential inflection points that may better capture gradients of rurality in relation to health outcomes. Moreover, the conservative thresholds applied to the healthcare access and utilization survey items increase specificity but may reduce comparability with studies that employ more inclusive criteria.

## Conclusions

In this study, we developed a standardized approach to identify and characterize rural participation and enrollment based on the 3-digit ZIP code geolocation function within the *All of Us* Researcher Workbench. Our study findings indicate that higher rurality is significantly associated with increased reports of delayed care and difficulty with healthcare affordability among *All of Us* Participants. This work highlights both the potential and current gaps in understanding rural health within

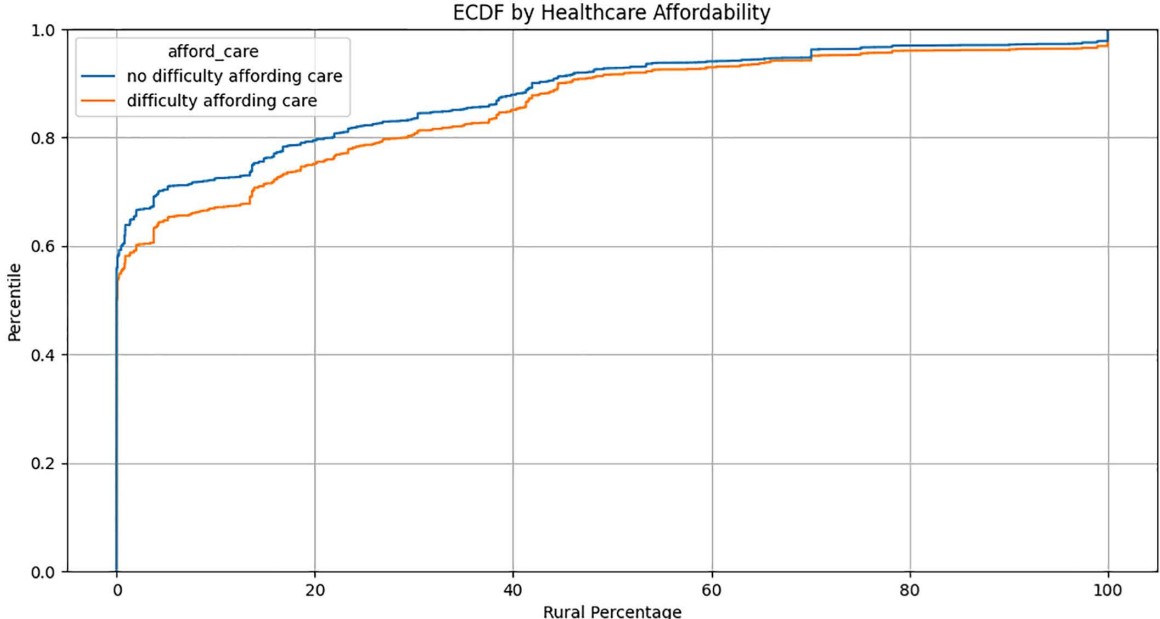

**Fig 5. ECDF Plot showing the cumulative distribution of Healthcare Affordability along the rurality scale.** At 0% rural, we observe that the two curves uniformly rise steeply, indicating that a significant proportion of the participants have similar levels of healthcare affordability. The two curves then split at different percentiles, indicating a disparity in healthcare affordability with respect to the rural percentage. The orange curve (difficulty affording care) indicates that as the percentile increases, those experiencing difficulty affording care tend to come from higher rural percentages when compared to those with no difficulty affording care (blue curve).

large-scale observational health datasets by characterizing rural participation and examining associations between rurality, delayed care, and healthcare affordability. The rurality scale proposed in this study has the potential to expand both the scope and quality of rurality-focused research within the *All of Us* Researcher Workbench. This work represents an important first step in quantifying rural representation within *All of Us* and provides the foundation for future investigations, not only into how rurality intersects with social determinants of health and other health outcomes, but also for developing targeted interventions to improve health outcomes and reduce health disparities in rural America.

## Supporting information

**S1 Fig. Jitter plot illustrating the distribution of three racial groups across varying rural percentages.** There is a decrease in racial diversity with increasing rural percentages.
(TIF)

## Acknowledgments

We gratefully acknowledge *All of Us* participants for their contributions, without whom this research would not have been possible. We also thank the National Institutes of Health's *All of Us* Research Program for making available the participant data examined in this study. This study used data from the *All of Us* Research Program's Controlled Tier Dataset version 8, available to authorized users on the Researcher Workbench.

Our project team includes a community scientist-clinician (MB), an informatician (VS), and an informatics trainee (TO). MB is a physician champion for the *All of Us* Research Program at the University of Arizona and Banner Health and Vice-Chief of Staff at Banner North Colorado Medical Center in Greeley, Colorado. In these roles, he advocates at the local and

national levels for the ability to study rural health outcomes within the *All of Us* dataset. VS is a health systems scientist and co-lead for researcher engagement with the *All of Us* program. This project grew out of MB's keen interest in rural health and research related to rural health outcomes.

## Author contributions

**Conceptualization:** Michael Bradfield.

**Data curation:** Toluwanimi Olorunnisola.

**Formal analysis:** Toluwanimi Olorunnisola.

**Funding acquisition:** Michael Bradfield, Vignesh Subbian.

**Investigation:** Toluwanimi Olorunnisola, Michael Bradfield.

**Methodology:** Toluwanimi Olorunnisola, Vignesh Subbian.

**Project administration:** Toluwanimi Olorunnisola, Michael Bradfield, Vignesh Subbian.

**Software:** Toluwanimi Olorunnisola.

**Supervision:** Michael Bradfield, Vignesh Subbian.

**Validation:** Toluwanimi Olorunnisola, Vignesh Subbian.

**Visualization:** Toluwanimi Olorunnisola, Vignesh Subbian.

**Writing – original draft:** Toluwanimi Olorunnisola, Michael Bradfield, Vignesh Subbian.

**Writing – review & editing:** Toluwanimi Olorunnisola, Michael Bradfield, Vignesh Subbian.

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
