## [Decision Letter · Decision Letter 0]

24 Aug 2025

Dear Dr. Olorunnisola,

Thank you for submitting your manuscript to PLOS ONE. After careful consideration, we feel that it has merit but does not fully meet PLOS ONE’s publication criteria as it currently stands. Therefore, we invite you to submit a revised version of the manuscript that addresses the points raised during the review process.

We look forward to receiving your revised manuscript.

Kind regards,

Taiwo Opeyemi Aremu, MD, MPH, PhD

Academic Editor

PLOS ONE

 [The All of Us Research Program is supported by the National Institutes of Health, Office of the Director, including grants OT2OD026549 and OT2OD036485.].

3. Thank you for uploading your study's underlying data set. Unfortunately, the repository you have noted in your Data Availability statement does not qualify as an acceptable data repository according to PLOS's standards.

4. We note that Figure 3 in your submission contain [map/satellite] images which may be copyrighted. All PLOS content is published under the Creative Commons Attribution License (CC BY 4.0), which means that the manuscript, images, and Supporting Information files will be freely available online, and any third party is permitted to access, download, copy, distribute, and use these materials in any way, even commercially, with proper attribution. For these reasons, we cannot publish previously copyrighted maps or satellite images created using proprietary data, such as Google software (Google Maps, Street View, and Earth). For more information, see our copyright guidelines: http://journals.plos.org/plosone/s/licenses-and-copyright.

1. You may seek permission from the original copyright holder of Figure 3 to publish the content specifically under the CC BY 4.0 license.   

Additional Editor Comments (if provided):

Reviewers' comments:

Reviewer's Responses to Questions

**Comments to the Author**

1. Is the manuscript technically sound, and do the data support the conclusions?

Reviewer #1: Yes

Reviewer #2: Yes

2. Has the statistical analysis been performed appropriately and rigorously?

Reviewer #1: Yes

Reviewer #2: Yes

3. Have the authors made all data underlying the findings in their manuscript fully available?

Reviewer #1: Yes

Reviewer #2: Yes

4. Is the manuscript presented in an intelligible fashion and written in standard English?

Reviewer #1: Yes

Reviewer #2: Yes

Reviewer #1: Thank you for the opportunity to review this important research. This is a well-prepared manuscript that addresses an important methodological issue in rural health research: the lack of consistent and privacy-compliant measures of rurality in large-scale datasets. The study’s strengths include its use of a very large, diverse cohort (All of Us), transparency in method development, and focus on operationalizing a practical rurality scale.

I present some oppportunities for improvement:

Provide justification or sensitivity testing for the thresholds used in survey coding.

Expand discussion comparing this rurality scale with existing measures (e.g., RUCA, IRR, county-level designations).

Consider additional analyses (e.g., logistic regression) to test associations while controlling for confounders.

Clarify limitations related to the use of 3-digit ZIP codes and potential ecological bias.

Expand figure captions to allow interpretation without reference to the main text.

Proofread for minor grammatical and stylistic improvements.

Reviewer #2: Comments

Introduction section

The introduction moves logically from the general problem of rural health to the specific challenge of defining rurality, to the limitations of the All of Us dataset, and finally, to the study's proposed solution. However, while the paper clearly states the purpose, the final sentences of the introduction could be strengthened to more explicitly highlight the broader implications of the work. For example, in addition to informing future research, how might this rurality scale be used by policymakers or public health organizations? Also, the introduction uses several citations to support its claims. You could consider starting the introduction with a single, compelling statistic about the life expectancy gap between rural and urban populations to immediately grab the reader's attention and underscore the urgency of the problem.

Methods

This is a well-structured and detailed "Methods" section that clearly outlines the study's approach. It effectively explains the different tiers of the All of Us dataset and, most importantly, provides a transparent, step-by-step process for creating the rurality scale. To further strengthen the methods section, there should be a justification for the threshold. For instance, in the section on "Healthcare access and utilization survey data preparation and analysis," the authors state that thresholds of "six or more affirmative responses" for delayed care and "nine or more affirmative responses" for affordability were selected to reflect a "high burden." This is a critical point that needs more robust justification. Authors should explain the rationale behind these specific numbers. Were they based on a pre-existing clinical standard, a statistical analysis (e.g., a percentile cutoff), or a consensus among the research team? Providing a clearer justification will strengthen the validity of the findings.

There must be consistency in Language: The term "rurality" is used, but the scale is also referred to as a "rural-urban continuum" in the figure caption. While these are related, using consistent terminology throughout the paper will prevent any potential confusion.

Discussion Section

The discussion correctly re-establishes the context by referencing the existing literature on rurality definitions. It effectively highlights the study's unique methodological strength—the creation of a continuous rurality scale compatible with de-identified data. However, here is a suggestion for revision:

Consider reorganizing the discussion to maximize its impact. A potential flow could be:

1. Summary of Key Findings: Start with a concise summary of your most significant findings from the results section. For example, "This study shows that participants living in areas with higher rurality percentages, as defined by our novel scale, report statistically significant disparities in delayed care and healthcare affordability compared to their urban counterparts."

2. Connection to Literature and Contribution: Next, explain how these findings align with or contradict existing research. Reiterate your study's unique contribution—the development of a practical rurality scale—in this context.

3. Limitations and Implications: Acknowledge any limitations of your methodology (e.g., reliance on 3-digit ZIP codes). Then, discuss the broader implications of your findings for patient care, policy, and research.

4. Future Work: Conclude by providing a clear and detailed roadmap for future research using your developed scale.

Conclusion

This is a strong conclusion that effectively summarizes the study's key contributions. It brings the paper full circle by re-emphasizing the problem and presenting the proposed rurality scale as a viable solution.

Critique of the Conclusions Section

• Strengths: The conclusion is concise and effectively restates the study's main purpose and findings. It correctly identifies the potential of the rurality scale to expand the scope and quality of research within the All of Us program.

• Areas for Improvement:

o Specificity: The conclusion mentions "potential and current gaps" and "associations between rurality, delayed care, and healthcare affordability" without directly stating the findings. It would be stronger to explicitly state what was found.

o Proposed Revision: Start with a more direct summary of the results. "In this study, we developed a standardized approach to identify and characterize rural participants within the All of Us program and found that higher rurality was significantly associated with increased reports of delayed care and difficulty with healthcare affordability."

o "So What?" Moment: The last sentence is good, but it can be enhanced to leave a lasting impression. What is the ultimate goal of this work? It's not just about "quantifying representation"—it's about improving health equity.

o Proposed Revision: The final sentence could be more ambitious. This work provides a critical foundation for future investigations, not only into how rurality intersects with social determinants of health but also for developing targeted interventions to improve health outcomes and reduce health disparities in rural America.

**Do you want your identity to be public for this peer review?** For information about this choice, including consent withdrawal, please see our Privacy Policy

Reviewer #1: **Yes: ** Noriel Calaguas

Reviewer #2: No

---

## [Author Response · Author response to Decision Letter 1]

23 Sep 2025

Thank you for giving us the opportunity to revise our manuscript entitled “Characterizing rurality using the All of Us research program data” (Manuscript ID: PONE-D-25-36901). We sincerely appreciate the time and effort the reviewers and editors have invested in providing constructive feedback, which has helped us improve the quality and clarity of our work. We have carefully considered each comment and revised the manuscript accordingly. We have provided a detailed, point-by-point response to all reviewers’ comments. For clarity, we have reproduced each comment and followed it with our response. All changes are highlighted in the “Revised Manuscript with Track Changes” document.

We have also addressed the comments raised regarding adherence to the journal requirements in the rebuttal letter (Response to Reviewers).

Thank you for your time and consideration of our manuscript. We look forward to hearing from you soon.

---

## [Decision Letter · Decision Letter 1]

6 Oct 2025

Characterizing rurality using the All of Us research program data

PONE-D-25-36901R1

Dear Dr. Olorunnisola,

We’re pleased to inform you that your manuscript has been judged scientifically suitable for publication and will be formally accepted for publication once it meets all outstanding technical requirements.

Kind regards,

Taiwo Opeyemi Aremu, MD, MPH, PhD

Academic Editor

PLOS ONE

Additional Editor Comments (optional):

Reviewers' comments:

Reviewer's Responses to Questions

**Comments to the Author**

Reviewer #1: All comments have been addressed

2. Is the manuscript technically sound, and do the data support the conclusions?

Reviewer #1: (No Response)

3. Has the statistical analysis been performed appropriately and rigorously?

Reviewer #1: (No Response)

4. Have the authors made all data underlying the findings in their manuscript fully available?

Reviewer #1: (No Response)

5. Is the manuscript presented in an intelligible fashion and written in standard English?

Reviewer #1: (No Response)

Reviewer #1: (No Response)

**Do you want your identity to be public for this peer review?** For information about this choice, including consent withdrawal, please see our Privacy Policy

Reviewer #1: **Yes: ** Noriel P. Calaguas, PhD, MSHSA, RN, ACRN

---

## [Editor Report · Acceptance letter]

PONE-D-25-36901R1

PLOS ONE

Dear Dr. Olorunnisola,

I'm pleased to inform you that your manuscript has been deemed suitable for publication in PLOS ONE. Congratulations! Your manuscript is now being handed over to our production team.

Kind regards,

on behalf of

Dr. Taiwo Opeyemi Aremu

Academic Editor

PLOS ONE